# Effects of Freezing Raw Yak Milk on the Fermentation Performance and Storage Quality of Yogurt

**DOI:** 10.3390/foods12173223

**Published:** 2023-08-27

**Authors:** Aili Li, Xueting Han, Jie Zheng, Jianing Zhai, Nan Cui, Peng Du, Jian Xu

**Affiliations:** Key Laboratory of Dairy Science, Ministry of Education, College of Food Science, Northeast Agricultural University, No. 600, Changjiang Road, Harbin 150030, China; ailimail@neau.edu.cn (A.L.); mailhxt@163.com (X.H.); s211001015@neau.edu.cn (J.Z.); 18145624219@163.com (J.Z.); nancuinan@163.com (N.C.); dupeng@neau.edu.cn (P.D.)

**Keywords:** yak milk, yak yogurt, frozen, storage, fermentation

## Abstract

In this study, the effects of freezing yak milk at −20 °C and −40 °C for 30, 90 and 180 days on the fermentation characteristics and storage quality of the corresponding yogurt were discussed. The results showed that, compared with that of yogurt made from fresh yak milk, the lactic acid bacteria (LAB) growth and acid production rate of the yogurt in the −20 °C group decreased at 90 d. The water-holding capacity, viscosity and hardness decreased during storage, and a sour taste was prominent, while no significant changes were observed in the −40 °C group. At 180 d of freezing, the post-acidification of the yogurt in the −20 °C and −40 °C groups increased after 21 d of storage. Compared with the −40 °C group, the −20 °C group showed a significant decrease in LAB counts, a decrease in pH value to 3.63–3.80 and poor texture and sensory quality.

## 1. Introduction

Yak milk is a high-quality milk source unique to the Tibetan Plateau region of China. The yogurt produced from it not only has a better nutritional composition than ordinary commercially available yogurt but also has a unique flavor and taste, as well as a variety of health functions, such as scavenging free radicals [1], lowering cholesterol [2,3] and strengthening immunity [4]. The quality of yogurt is directly affected by the quality of the raw milk. Due to the harsh climate in highland areas, the annual lactation period of yaks is only 105–180 days, and their annual milk production is only about 5% of that of Holstein cows [5,6]. In addition, due to difficulties in milking, collecting milk and transporting, local herders often use traditional refrigerator freezing at −20 °C to extend the shelf life of raw yak milk, alleviating regional and seasonal shortages of fresh milk [7]. However, the effect of frozen storage on the quality of raw yak milk is not clear so far, which has led to unstable and non-standardized yak yogurt quality in Tibetan areas. This has become a key problem that has limited the development of yak dairy products.

Freezing, an important storage method, effectively controls the propagation of harmful microorganisms in milk. However, during freezing, a phase change will occur, and the salt, sugar and organic acids in the milk can concentrate, leading to protein salting out and lactose crystallization [8]. Among these effects, the conversion of β-lactose to the less soluble α-lactose hydrate form will cause protein dehydration and lead to larger ice crystal growth. The resulting increases in ionic strength and osmotic pressure in the milk will also decrease the structural stability of casein colloids, and this phenomenon will increase with storage time [9]. Protein salting and lactose crystallization also directly affect the carbon-to-nitrogen ratio and the content required for the growth of lactic acid bacteria (LAB), as well as the utilization of carbon and nitrogen sources by it. In addition, the high content of microorganisms in raw milk can compete with the growth of LAB. Among them, psychrotrophic bacteria are the dominant bacteria in the metabolic growth of raw milk during low-temperature storage [10,11]; for example, the *Pseudomonas* spp. produces peptidases that hydrolyze casein and cause the gradual formation of particles and sediments in raw milk during storage [12]. These are also common risk factors for poor curdling or non-curdling during cheese processing [13]. Currently, only a few studies have evaluated the effects of long-term refrigeration/freezing on goat and sheep milk and the consequences for the yogurt and cheese produced from these types of milk [14,15]. It is necessary to explore the effects of the frozen storage of raw yak milk on the subsequent yogurt production.

Our research team has carried out some work on improving the quality and shelf life of yak yogurt based on a project of the Key Technology Research and Product Development for Ecological Livestock Products Processing on Qinghai-Tibet Plateau [16,17]. We have further investigated the effects of different raw milk freezing conditions on fermentation performance and the post-acidification, texture and sensory characteristics of yak yogurt based on the current production status of the Yak Dairy Products Processing Cooperative in Yushu Autonomous Prefecture, Qinghai, where raw milk is frozen for a long time, and high-temperature instantaneous sterilization equipment is lacking. This study provides some basic theoretical parameters for the quality control of yak milk and its fermented dairy products during processing and storage. Also, it is of practical significance to improve the processing level of fermented yak dairy products in Tibetan areas and increase the economic benefits for herders.

## 2. Materials and Methods

### 2.1. Raw Yak Milk

Raw yak milk was provided by the Yushu Tibetan Autonomous Prefecture, Qinghai Province, from the same batch of fresh yak milk in the middle lactation period and conformed to the RHB801-2012-Raw Yak Milk Industry Specification (fat content: 6.20 ± 0.02%; protein content: 4.66 ± 0.06%; nonfat milk solids: 9.00 ± 1.09%; dry matter content: 18.09 ± 0.30%; ash content: 0.80 ± 0.02%; lactose content: 5.41 ± 0.35%). Samples were collected on the farm, frozen at −80 °C, flown to the laboratory and thawed in water at room temperature (25 °C). No collected raw milk was subjected to a sterilization operation, and the raw milk was packed in aseptic bags (100 mL each) and frozen in a refrigerator at −20 °C and −40 °C for 30, 90 and 180 days, respectively.

### 2.2. Composition, pH and Acidity of Raw Milk

Fresh and frozen raw yak milk was characterized according to the AOAC method [18] for protein (micro-Kjeldahl method (K9840, Shanghai, China, method 991.20) and fat (Gerber [THR16A, Changsha, China, method 2000.18])). pH values were determined using a digital pH meter (Model PHS-3C, Shanghai, China), and acidity was determined according to the AOAC method (method 947.05).

### 2.3. Protein Hydrolysis Degrees of Raw Milk

The degrees of protein hydrolysis (DH) of the raw milk were measured with the o-phthalaldehyde (OPA) method, according to the method of Mulcahy et al. [19] but with slight modifications. Briefly, the absorbances of the samples at 340 nm were measured with a UV-visible spectrophotometer (UV-2401 PC UV spectrophotometer, Shimadzu Laboratory Equipment Co., Ltd., Shanghai, China) using serine (0–0.1 mg/mL) as a standard protein. Quantification of the amino groups was performed by referring to the serine standard curve with the absorbance at 340 nm (y = 0.2843x + 0.1271, R^2^ = 0.9993).

### 2.4. Microbiological Characterization of Raw Milk

Fresh and frozen raw milk was measured using the plate colony counting method for total plate count (TPC) and total psychrophilic count (TPsC) [20].

### 2.5. Thermal Stability of Raw Milk

Referring to the method of Crowley et al. [21], the test tube containing 1 mL of raw yak milk was placed in an oil bath at 140 °C at room temperature (25 °C). The thermal stability (TS) of the milk was assessed based on the time required from placement to the start of hanging flocculation and precipitation.

### 2.6. Determination of Fat Oxidation of Raw Milk (TBRAS)

Thus, 1 mL of yak raw milk, 3 mL of 0.02 mol/L thiobarbituric acid solution and 17 mL of trichloroacetic acid-ethylenediaminetetraacetic acid (TCA-EDTA) solution were placed in a 50 mL centrifuge tube, mixed, boiled for 30 min and cooled to room temperature. Then, 4 mL supernatant was mixed with an equal volume of chloroform and centrifuged at 3000 r/min for 15 min. The absorbance of the supernatant was measured at 532 nm and 600 nm, respectively. Distilled water was used as a control. The TBRAS value was expressed in milligrams of malondialdehyde per kilogram of lipid oxidation sample solution (Equation (1)).
(1)$$\mathrm{TBRAS}=\left(A_{532}-{\;A}_{600}\right)\;\times \;9.48$$
where A_532_ and A_600_ represent the absorbance values of the solution at 532 and 600 nm, respectively, and 9.48 is a constant.

### 2.7. Preparation of Yak Yogurt

Raw yak milk frozen at −20 °C and −40 °C for 30, 90 and 180 days was thawed (25 °C), and the same batch of fresh raw yak milk was used as a control group for yogurt production. Each group of samples was pasteurized (62 °C, 30 min) and homogenized (15 MPa). When the temperature dropped to 43 °C, 10^6^ CFU/mL of yogurt starter MY105 (containing *Lactobacillus bulgaricus* and *Streptococcus thermophilus* in a 1:1 ratio, Danisco (Jiangsu, China) Ltd.) was added to the yak milk at 2% inoculation rate. All experimental groups were prepared with 3 copies, a total of 21 replicates and fermented at 43 °C until pH dropped to 4.5. After fermentation, the yogurt was stored at 4 °C for 21 d for subsequent experiments.

### 2.8. pH and Acidity of Yogurt

The pH and acidity of the yogurt were measured as described above. Specifically, the pH of the yogurt was measured every 30 min during the fermentation process until the pH dropped to 4.5. The fermented yogurt was refrigerated at 4 °C and evaluated at 1 and 21 days of storage. The results obtained were simulated using the modified Gompertz equation [22] of bacterial growth applicable to the pH drop during fermentation (Equation (2)).
(2)$$\mathrm{pH}={\mathrm{pH}}_{0}+(\mathrm{pH}\infty -\;{\mathrm{pH}}_{0})\exp\{-\exp[(\lambda \;-\;t)\mu \;e/(\mathrm{pH}\infty -\;{\mathrm{pH}}_{0})+1]\}$$
where pH_0_ = initial pH, pH_∞_ = final pH, μ = maximum pH reduction rate (h^−1^), λ = lag phase time (h) and t = time (h).

### 2.9. LAB Counts of Yogurt

The LAB counts were measured during fermentation and storage (1 d and 21 d). The *Lb. bulgaricus* and *S. thermophilus* were counted using MRS and M17 solid medium, respectively. All media were incubated at 37 °C for 72 h. The maximum growth rates V_max Lb_ and V_max St_ were determined after measuring the growth curve and were derived from the maximum slope fitted to the logarithmic growth phase curve.

### 2.10. Determination of Water-Holding Capacity of Yogurt

The water-holding capacity (WHC) of yogurt was measured after 24 h of production and on the 21st day of storage. The mass of the centrifuge tube was weighed m_0_ (g), the 5–8 mL of yogurt was added to the tube, the total mass was weighed as m_1_ (g), the supernatant was removed after centrifugation at 4000 r/min for 20 min and the remaining mass was m_2_ (g). The WHC was converted according to the following equation (Equation (3)).
(3)$$\mathrm{WHC}\;\left(\%\right)=\frac{m_{2}-m_{0}}{m_{1}-m_{0}}\;\times \;100\%$$

### 2.11. Texture Profile Analysis of Yogurt

Yogurt samples (30 mL) were dispensed into plastic containers immediately after preparation and post-matured at 4 °C for 24 h. The texture profile analysis (TPA) of yak yogurt was carried out using a TA-300W texture analyzer (Shanghai, China). The measurement conditions were as follows: probe P/25, pre-test speed of 4 mm/s, test speed of 1 mm/s, post-test speed of 2 mm/s and trigger force of 5.0 g. After the measurement, the hardness (g) and adhesiveness (g*s) were calculated and analyzed automatically using the dedicated software Exponent. The same operation was performed on the 21st day of yogurt storage. Three replicates of each sample were tested.

### 2.12. Rheological Properties of Yogurt

According to the method of Zhang et al. [17], the rheological properties of yogurt were tested with the HR-1 rotational rheometer (TA discovery rotational rheometer HR1, New Castle, DE, USA). Before testing, the yogurt samples were rested at 4 °C for 10 min. The samples were used for static shear scans and dynamic frequency scans after being evenly distributed. Static shear scan: shear rates from 0.1 s^−1^ to 300.0 s^−1^ were used to determine the apparent viscosity of yogurt as a function of shear rate. Dynamic frequency scan: oscillation strain of 0.5% and frequency scan from 0.01 to 100 Hz with a fixed frequency of 1 Hz to test the elasticity modulus (G′) and viscous modulus (G″) of yogurt.

### 2.13. Scanning Electron Microscope Observation of Yogurt

Small strips of 2 × 5 mm were cut from the center of yogurt stored for 21 d and fixed in 2.5% glutaraldehyde solution (pH 6.8) for 6 h. After fixation, the samples were rinsed three times with phosphate buffer (pH 6.8) and then dehydrated with 50%, 70% and 90% ethanol, respectively, and twice with 100% ethanol for 10 min each. The samples were frozen at −20 °C for 30 min and dried in a freeze dryer (ES-2030, Hitachi, Japan), and the dried samples were fixed on a sample table with tape and coated with an ion sputter coater. The samples were observed and photographed using a scanning electron microscope (SEM) S-3400 (S-3400, Hitachi, Japan).

### 2.14. Sensory Evaluation of Yogurt

The sensory characteristics of yogurt consisted of two-part experiments. First, we put 35 mL of diluted yak yogurt into an electronic tongue sampling cup, sealed and equilibrated at room temperature for 20 min and measured with an SA-402B-type electronic tongue. The sensor types were AE1 for astringent taste, C00 for bitter taste, GL1 for sweet taste, AAE for salty taste and CAO for sour taste, and they were subjected to principal component analysis (PCA).

Next, we invited 20 trained judges to evaluate the sensory characteristics of the yogurt (ambient temperature of 25 °C and sample temperature of 4 °C). This type of research does not require formal ethical approval in China. All participants were informed of the purpose of the study and completed an informed consent form. The sensory panel consisted of 10 males and 10 females, aged between 18 and 35 years, trained five times according to the Chinese national standard (GB/T 16291.1-2012: Sensory analysis—General guidance for the selection, training and monitoring of assessors). All yogurt samples were placed in individual plastic bottles, coded with random numbers and provided to the panel members. Each member received a 10-point scale for each characteristic (texture, sourness flavor, appearance and overall acceptability) according to the method described by Heydari et al. [23], where 0 = “very poor” and 10 = “excellent”. The results are given as the average of three trials for each sample and form a radar plot.

### 2.15. Statistical Analysis

All data were measured three times and statistically analyzed using Excel (2019) software. SPSS 26 was used for analysis of variance (ANOVA), Duncan’s multiple comparisons and significance analysis (α confidence level of 95%). Origin 2019 was used to perform principal component analysis and plot the images. All test data were expressed as mean ± standard deviation.

## 3. Results and Discussion

### 3.1. Effects of Frozen Storage on the Quality of Raw Yak Milk

We first analyzed the changes in the quality of raw yak milk under different freezing conditions (−20 °C/−40 °C, 30 d/90 d/180 d). As shown in Table 1, the TS, FOD, acidity and dry matter content of the raw milk frozen at −20 °C and −40 °C for 30 d did not change significantly compared to the FM group (*p* > 0.05). The quality of the milk frozen for 90 d decreased, and the protein content, pH and TS of the milk in the −20 °C group were significantly lower than that in the −40 °C group, FOD was significantly higher than the −40 °C group (*p* < 0.05), but their physicochemical indexes were still within the range of RHB801 “Raw Yak Milk”. When frozen up to 180 d, the protein content of the −20 °C group decreased to 3.59 ± 0.09%, and the acidity increased to 2.23 ± 0.02%, which slightly exceeded the production standard (protein content ≥ 3.8%). However, the raw milk in the −40 °C group still had a high dry matter content, and all indexes were in line with the production standard. We also found that the TPC and TPsC of raw milk stored at −20 °C for 30 d, 90 d and 180 d groups were generally higher than those of the −40 °C group (*p* < 0.05) (Table 1). This may be due to the increased concentration of local solutes during freezing, leading to the dehydration of microbial cells; the lower freezing rate as well as the larger ice crystal nuclei reduce the survival of microorganisms [8,24]. During the freezing process, the growth and metabolism of psychrophilic bacteria secreted lipase and protease, which degraded the protein and fat in raw milk. For example, Meng et al. [25] identified *Pseudomonas* spp. isolated from goat, buffalo, camel and yak milk, where the yak milk isolates exhibited high proteolytic activity at low temperatures. We further analyzed the composition and function of psychrophilic bacteria in frozen yak milk and found a significant positive correlation with protein hydrolysis and fat oxidation (data not shown). In addition, mechanical damage to the fat globule membrane from larger ice crystals during freezing also facilitates lipase catabolism and the transfer of fat globule membrane proteins [8,24]. In addition, microorganisms in frozen milk will continue to convert lactose into lactic acid, making the colloidal calcium phosphate partially dissolved, reducing the electrostatic repulsion between protein molecules and causing aggregation of proteins, thus reducing raw milk stability [8].

Similar to our results, Garcia et al. [26] found a significant decrease in the fat and caloric content of breast milk after 90 d of freezing at −20 °C. Yu et al. [27] found that the fat, protein and lactose content of goat milk decreased after 80 d of freezing and that fat loss was less in the −80 °C group compared to the −20 °C group. Qu et al. [24] and Zhang et al. [28] observed that breast milk frozen at −60 °C was more effective in maintaining the digestive properties of proteins, as well as active protein concentrations (e.g., immunoglobulins, lactoferrin, lysozyme), most similar to fresh breast milk compared to −18 °C. In addition, another study found that the composition content of goat milk was only slightly altered (*p* > 0.05) within 90 d of freezing at −30 °C [29]. This is closely related to the type and number of initial microorganisms in the milk and the rate of freezing/thawing [8,30]. Nevertheless, freezing has important effects on both compositional and textural changes in raw milk.

### 3.2. Effects of Raw Milk Frozen Storage on the Fermentation Performance of Yak Yogurt

Subsequently, we explored the status of frozen raw milk fermented yogurt. During yogurt fermentation, the pH dropped from 5.7 to 5.0 in the exponential phase of symbiotic growth of *Lb. bulgaricus* and *S. thermophilus*, which is important for LAB growth and acid production. The data we obtained fitted well (R^2^ ≥ 0.99) with the modified Gompertz model. This model was used to indirectly assess bacterial growth through the decrease in pH during fermentation, and the acidification curve is shown in Figure 1C. In Table 2, the parameter µ represented the maximum rate of decrease in pH and was calculated automatically based on the fitted equation. The results showed that there was no significant difference between the acid production rate of LAB in the 30 d frozen milk and FM group (*p* > 0.05). By 90 d of freezing, the acid production rate of LAB in the −20 °C and −40 °C groups was significantly lower than that in the FM group (−0.80 ± 0.11 h^−1^). By 180 d of freezing, the µ values of LAB in −20 °C and −40 °C groups decreased to −0.50 ± 0.04 h^−1^ and −0.53 ± 0.06 h^−1^, respectively, while the curdling time was extended to 6.5 h and 6 h. Correspondingly, the growth rates of LAB in the −20 °C and −40 °C groups were lower than those in the FM group after 90 d of freezing (Table 2). In particular, V_max St_ and V_max Lb_ were lowest in the −20 °C 180 d group, with counts of 8.50 ± 0.08 log CFU/mL and 7.51 ± 0.06 log CFU/mL, respectively, by the end of curdling (Figure 1A,B). The acid production rate and LAB growth rate were slightly higher in the −40 °C group than in the −20 °C group throughout the fermentation process. Similarly, Pazzola et al. [31] investigated the effects of freezing sheep milk at −20 °C for 1, 3 and 5 months on the quality of the cheese produced and found that, with longer freezing time, the fermentation time of sheep milk cheese increased, and the yield and quality decreased significantly. Tribst et al. [32,33] also found that in the application of frozen sheep milk for yogurt processing, the entry of LAB into the logarithmic growth period was delayed, and the maximum acidification rate was slowed down. This may be related to the reduction in raw milk nutrients (e.g., protein, lactose, etc.) caused by frozen storage. In addition, free fatty acids (such as linoleic acid, oleic acid, etc.) released by lipase during frozen storage also have an effect on the growth of LAB, which is not conducive to the acidification of LAB [34,35].

### 3.3. Effects of Frozen Raw Milk on the Storage Quality of Yak Yogurt

We then analyzed the effect of raw milk freezing on post-acidification, texture and sensory evaluation of yak yogurt over 21 d storage.

#### 3.3.1. Analysis of Post-Acidification Degree of Yak Yogurt

As shown in Table 3, after 21 d of storage, the acidity and microbiological indexes of yogurt in each group differed significantly (*p* < 0.05) from those at 1 d of storage. Compared with the FM group, there was no significant change in the 30 d group (*p* > 0.05), the acidity of yogurt increased and the LAB counts decreased in the −20 °C 90 d group (*p* < 0.05) and there was still no significant change in the −40 °C 90 d group. At 180 d of raw milk storage, the acidity of yogurt in the −20 °C group reached 12.24 ± 0.09%, and the number of *S. thermophilus* and *Lb. bulgaricus* decreased to 5.77 ± 0.06 log CFU/mL and 5.45 ± 0.10 log CFU/mL, respectively. Similarly, Tribst et al. [32] found that yogurt made from sheep milk frozen at −18 °C for 1 month showed a faster pH drop over 28 d of storage than fresh milk, with no significant decrease in LAB counts (> 8.69 log CFU/mL). Studies have shown that yogurt experiences a simultaneous drop in pH and LAB counts over 21 days of storage [36,37]. Our results showed that the LAB counts in yogurt in the FM group decreased by 1 log CFU/mL after 21 days of storage; however, the decrease was greater in the other groups, especially in the −20 °C 180 d group (< 6 log CFU/mL). We hypothesized that this may be related to the fact that the quality of raw milk was more severely impaired by prolonged storage. The high buffering capacity can counteract the damage caused by excessive acidification to LAB by slowing down the rate of pH drop, thus maintaining the viable bacteria counts during storage of yogurt [32]. For example, sheep’s milk has a higher dry matter content than cow’s milk, and its buffering capacity is correspondingly higher, as well as its stability after long-term frozen storage. Moreover, the growth and acidification rate of *Lb. bulgaricus* in goat milk is faster and the peptidase activity is increased compared to that in cow milk [33,38]. Michale et al. [39] also found that yogurt supplemented with plant extracts improved its buffering capacity due to its increased total solids content, which slowed down its post-acidification.

#### 3.3.2. Analysis of the Texture of Yak Yogurt

Post-acidification leads to increased hydrophobic and electrostatic interactions between proteins, increased casein particle size, colloidal calcium phosphate dissolution and partial reorganization of the protein network, thus affecting the textural properties of the product [40]. We found that the elasticity modulus (G′), viscosity modulus (G″) and apparent viscosity of each group of yogurt decreased with increasing freezing time of raw milk (30 d, 90 d, 180 d) and/or storage time of yogurt (1 d, 21 d) (Figure 2 and Figure 3). As shown in Table 4, the hardness, adhesion and WHC of yogurt in the 30 d group were similar to those in the FM group at the beginning of storage (*p* > 0.05), while the textural properties were significantly different after 21 d of storage (*p* < 0.05). When raw milk was frozen up to 90 d and 180 d, the hardness and viscosity of yogurt were further reduced after storage, and there were significant differences between the −20 °C and −40 °C groups (*p* < 0.05). Similarly, Tribst et al. [33] found that yogurt made from sheep milk frozen at −20 °C for 24 h had a reduced viscosity compared to yogurt made from fresh milk. Subsequently, Tribst et al. [32] made yogurt from sheep milk frozen at −20 °C for 1 month and still found that the hardness and viscosity of the yogurt decreased.

During the freezing of raw milk, the formation of ice crystals and supersaturated salt solutions can weaken the electrostatic interactions between casein micelles in the yogurt gel, resulting in a loose structure [8]. Our SEM results also showed (Figure 4) that after 21 d of storage, compared with the FM group, the yogurt in the 30 d group had a small number of pores but still had a tight network structure. The 90 d group showed an increase in pores and a decrease in protein cross-linking. The 180 d group showed a looser microstructure, exhibiting more open structures and a further increase in pores, which was consistent with the decrease in WHC and viscosity index of the yogurt. We also found that the electron microscope images of yogurt in the −20 °C group showed a looser structure and larger pores than those in the −40 °C group. Correspondingly, after 21 d of storage, the rheological and textural indexes of yogurt in the −20 °C group were significantly lower than those in the −40 °C group. This may be because the faster the freezing, the lower the effect on milk protein is. Yu et al. [41] reported that the higher protein content increased the hardness and viscosity of yogurt with a stronger three-dimensional reticulation. Biegalski et al. [42] found that pasta filata cheese processed using goat milk frozen for about 90 d lost refreshment, elasticity and gloss.

#### 3.3.3. Sensory Evaluation of Yak Yogurt

Figure 5A shows the PCA results of the electronic tongue taste test after 21 d of yogurt storage. The PC1 was 56.6%, the PC2 was 23.1% and the cumulative contribution rate was 79.70%. The FM group and the freezing groups were distributed in four quadrants; among them, the FM group was closer to the 30 d group and further away from the 90 d and 180 d freezing groups.

Meanwhile, as shown in Figure 5B, the taste variation of yogurt in the −40 °C group was always smaller than that in the −20 °C group. Yogurt in the 30 d group was more similar to that in the FM group, and with the increase in raw milk freezing days, the fresh taste signals of yogurt made in the 90 d and 180 d groups were weakened, and the sour, bitter and bitter aftertaste signals were relatively enhanced. Particularly, the sourness and bitterness of the yogurt in the −20 °C 180 d group were the most obvious.

Figure 5C shows the sensory evaluation of the yogurt by the 20 panelists. Similar to the electronic tongue results, the texture, appearance, flavor and overall acceptability scores of yogurt decreased with the increasing storage time of raw milk, while the acidity scores showed the opposite trend. In particular, the acidity score of the yogurt in the −20 °C 180 d group was 9.9 points, which was significantly higher than that in the FM group (1.25 points), and cracks appeared on the surface of the yogurt, and whey precipitation was more serious. Similarly, Spitzer et al. [43] found that over 180 d of freezing at −19 °C, breast milk was subjected to oxidation, and the fatty acid composition was altered, resulting in the production of unpleasant odors, such as fishy, metallic and sour taste. Fonseca et al. [44] found that when sheep milk was refrigerated at 4 °C for 5 d, milk powder made from it had increased total free fatty acid content and prominent sourness over 180 d of storage, reducing consumer acceptability.

## 4. Conclusions

In this study, the effects of raw yak milk stored at −20 °C and −40 °C for 30, 90 and 180 d on the quality of yogurt processing were investigated. The results showed that freezing raw milk at −20 °C or −40 °C for 30 d had little effect on the quality of the yogurt produced. When the raw milk was frozen for 90 d, the quality of yogurt in the −40 °C group was acceptable, while that in the −20 °C group was significantly decreased. Storage for 180 d was not favorable for yogurt production, especially the yogurt in the −20 °C group, which had a lower bacteria count, more obvious post-acidification and the worst texture characteristics and sensory scores after 21 d of storage. The results of this study can provide a theoretical basis for the processing of yak milk after freezing and storing. A more comprehensive approach is still needed to further investigate the phenomena occurring in yak milk after freeze storage and to explore its effects on other yak milk products, such as cheese and milk powder.

## Figures and Tables

**Figure 1 foods-12-03223-f001:**
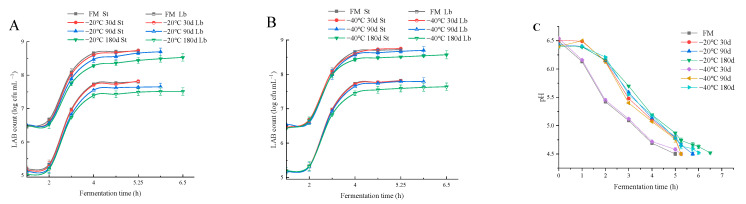
Variation of *Streptococcus thermophilus* (St) and *Lactobacillus bulgaricus* (Lb) populations with time during fermentation when raw milk frozen at −20 °C (**A**) and −40 °C (**B**) was used for yogurt production, and acidification curves (**C**). FM, fresh milk was used as control. Error lines represent the mean ± standard deviation of triplicate experiments.

**Figure 2 foods-12-03223-f002:**
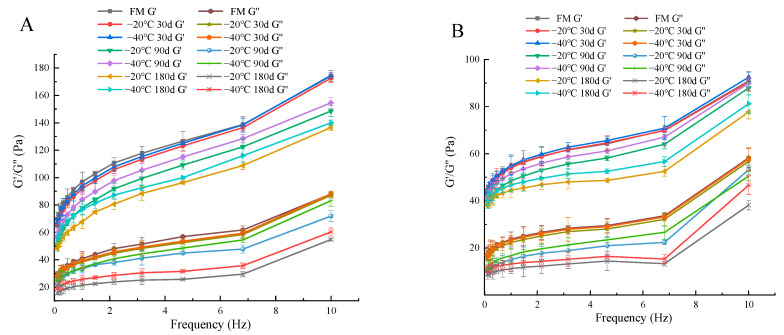
Variation of modulus with frequency of yak yogurt produced in different frozen groups (−20 °C, −40 °C; 30, 90 and 180 days) on day 1 (**A**) and day 21 (**B**) of storage. FM, fresh milk is the control group.

**Figure 3 foods-12-03223-f003:**
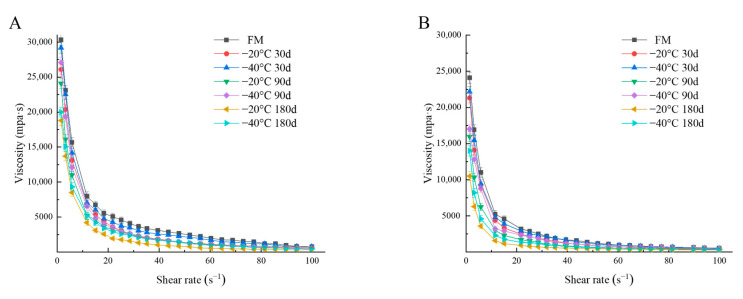
Variation of viscosity with shear rate of yak yogurt produced in different chilling groups (−20 °C, −40 °C; 30, 90 and 180 days) on day 1 (**A**) and day 21 (**B**) of storage. FM, fresh milk is the control group.

**Figure 4 foods-12-03223-f004:**
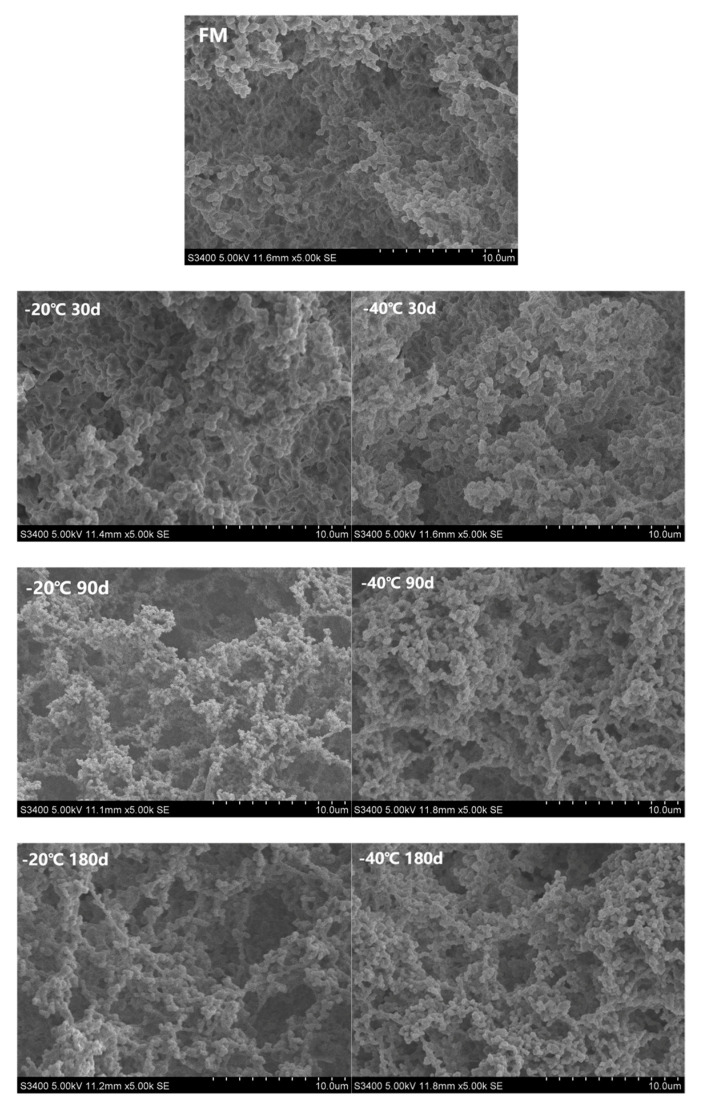
Microstructure of yak yogurt produced in different freezing groups (−20 °C, −40 °C; 30, 90 and 180 days) at day 21 of storage. Each figure was observed under 5000× electron microscope. FM, fresh milk.

**Figure 5 foods-12-03223-f005:**
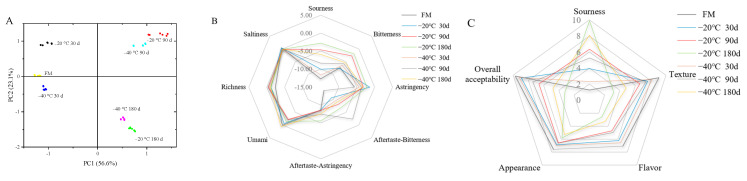
Sensory evaluation of yak yogurt produced in different chilled groups (−20 °C, −40 °C; 30, 90 and 180 days) at day 21 of storage, including electronic tongue taste test principal component analysis (PCA) (**A**) and electronic tongue taste radar plot (**B**), judge’s sensory evaluation radar plot (**C**). FM, fresh milk is the control group.

**Table 1 foods-12-03223-t001:** Effects of frozen storage on the physicochemical properties and freshness of raw yak milk.

Frozen Temperature	Frozen Time (Days)	Protein (%)	Fat (%)	DH (%)	FOD (mg/kg)	pH	Acidity (%)	TS (s)	TPC (log CFU/mL)	TPsC (log CFU/mL)
FM	-	4.66 ± 0.06 ^d^	6.20 ± 0.02 ^c^	2.27 ± 0.06 ^a^	0.009 ^a^	6.58 ± 0.02 ^d^	1.81 ± 0.02 ^a^	134.00 ± 4.16 ^e^	5.89 ± 0.02 ^e^	6.28 ± 0.03 ^e^
F (−20 °C)	30	4.58 ± 0.05 ^d^	6.12 ± 0.03 ^c^	2.38 ± 0.10 ^a^	0.022 ^b^	6.56 ± 0.02 ^d^	1.86 ± 0.01 ^a^	130.00 ± 3.60 ^e^	5.40 ± 0.01 ^b^	5.95 ± 0.02 ^bc^
	90	3.90 ± 0.05 ^b^	5.86 ± 0.08 ^b^	5.50 ± 0.00 ^c^	0.038 ^e^	6.45 ± 0.02 ^b^	2.09 ± 0.03 ^b^	105.33 ± 2.51 ^c^	5.53 ± 0.03 ^c^	6.08 ± 0.03 ^d^
	180	3.59 ± 0.09 ^a^	5.65 ± 0.05 ^a^	8.34 ± 0.10 ^e^	0.047 ^f^	6.40 ± 0.02 ^a^	2.23 ± 0.02 ^c^	78.17 ± 1.76 ^a^	5.60 ± 0.04 ^d^	6.12 ± 0.06 ^d^
F (−40 °C)	30	4.60 ± 0.05 ^d^	6.17 ± 0.01 ^c^	2.33 ± 0.13 ^a^	0.011 ^a^	6.56 ± 0.02 ^d^	1.80 ± 0.01 ^a^	133.33 ± 3.78 ^e^	5.30 ± 0.02 ^a^	5.76 ± 0.03 ^a^
	90	4.13 ± 0.03 ^c^	5.91 ± 0.06 ^b^	5.18 ± 0.08 ^b^	0.025 ^c^	6.50 ± 0.01 ^c^	1.90 ± 0.02 ^a^	116.66 ± 2.51 ^d^	5.42 ± 0.04 ^b^	5.92 ± 0.03 ^b^
	180	3.95 ± 0.04 ^b^	5.71 ± 0.09 ^a^	7.84 ± 0.10 ^d^	0.033 ^d^	6.46 ± 0.02 ^b^	2.07 ± 0.02 ^b^	98.00 ± 2.00 ^b^	5.45 ± 0.02 ^b^	5.99 ± 0.02 ^c^

Abbreviations are: FM, fresh milk; F, frozen milk; DH, degree of protein hydrolysis; FOD, Fat oxidation degree; TS, thermal stability; TPC, total plate count; TPsC, total psychrophilic count. ^a–f^ Lowercase superscripts indicate significant differences (*p* < 0.05) between the mean values of various indicators of raw milk in fresh and different frozen storage groups as assessed by Duncan’s multiple comparison method.

**Table 2 foods-12-03223-t002:** Effects of frozen raw milk storage on the fermentation performance of yak yogurt.

Frozen Temperature	Frozen Time (Days)	µ (h^−1^)	R^2^	T_pH4.5_ (h)	V_max Lb_ (log CFU mL^−1^ h^−1^)	V_max St_ (log CFU mL^−1^ h^−1^)
FM	-	−0.80 ± 0.11 ^b^	0.99	5.25 ± 0.25 ^a^	1.97 ± 0.05 ^b^	1.63 ± 0.04 ^b^
F (−20 °C)	30	−0.73 ± 0.08 ^b^	0.99	5.25 ± 0.25 ^a^	1.97 ± 0.05 ^b^	1.63 ± 0.06 ^b^
	90	−0.60 ± 0.07 ^a^	0.99	5.75 ± 0.17 ^cd^	1.90 ± 0.06 ^b^	1.52± 0.03 ^b^
	180	−0.50 ± 0.04 ^a^	0.99	6.50 ± 0.17 ^e^	1.72 ± 0.08 ^a^	1.35 ± 0.02 ^a^
F (−40 °C)	30	−0.74 ± 0.05 ^b^	0.99	5.25 ± 0.42 ^ab^	1.97 ± 0.04 ^b^	1.63 ± 0.03 ^b^
	90	−0.59 ± 0.05 ^a^	0.99	5.50 ± 0.25 ^bc^	1.90 ± 0.04 ^b^	1.61 ± 0.05 ^b^
	180	−0.53 ± 0.06 ^a^	0.99	6.00 ± 0.25 ^d^	1.65 ± 0.05 ^a^	1.43 ± 0.03 ^a^

Abbreviations are: FM: fresh milk. µ is the maximum acidification rate during yogurt fermentation; TpH4.5 is the time required for pH to reach 4.5; V_max Lb_ is the maximum growth rate of *Lactobacillus bulgaricus* during yogurt fermentation, and V_max St_ is the maximum growth rate of *Streptococcus thermophilus* during yogurt fermentation. ^a–e^ Lowercase superscripts indicate significant differences (*p* < 0.05) between the mean values of various indicators of raw milk in fresh and different frozen storage groups as assessed by Duncan’s multiple comparison method.

**Table 3 foods-12-03223-t003:** Effects of frozen storage of raw yak milk on post-acidification and LAB counts of yogurt.

Storage Time (Days)	Frozen Temperature	Frozen Time (Days)	pH	Acidity (%)	St (log CFU/mL)	Lb (log CFU/mL)
1	FM	-	4.37 ± 0.02 ^d^	8.37 ± 0.04 ^a^	8.81 ± 0.05 ^b^	8.08 ± 0.03 ^c^
	F (−20 °C)	30	4.34 ± 0.03 ^d^	8.42 ± 0.05 ^a^	8.84 ± 0.02 ^b^	8.07 ± 0.03 ^bc^
		90	4.24 ± 0.01 ^b^	9.17 ± 0.04 ^bc^	8.74 ± 0.12 ^b^	7.93 ± 0.11 ^b^
		180	4.15 ± 0.00 ^a^	9.57 ± 0.09 ^d^	8.44 ± 0.15 ^a^	7.53 ± 0.10 ^a^
	F (−40 °C)	30	4.34 ± 0.03 ^d^	8.46 ± 0.04 ^a^	8.83 ± 0.03 ^b^	8.07 ± 0.03 ^bc^
		90	4.36 ± 0.01 ^d^	8.46 ± 0.07 ^a^	8.78 ± 0.06 ^b^	7.96 ± 0.09 ^bc^
		180	4.29 ± 0.01 ^c^	8.90 ± 0.05 ^b^	8.79 ± 0.09 ^b^	7.99 ± 0.08 ^bc^
21	FM	-	4.16 ± 0.01 ^e^*	10.26 ± 0.05 ^a^*	7.19 ± 0.01 ^d^*	7.06 ± 0.15 ^e^*
	F (−20 °C)	30	4.16 ± 0.01 ^e^*	10.26 ± 0.04 ^a^*	7.17 ± 0.01 ^d^*	7.15 ± 0.14 ^e^*
		90	3.76 ± 0.02 ^b^*	11.61 ± 0.07 ^b^*	6.38 ± 0.03 ^b^*	6.05 ± 0.02 ^b^*
		180	3.63 ± 0.01 ^a^*	12.24 ± 0.09 ^c^*	5.77 ± 0.06 ^a^*	5.45 ± 0.10 ^a^*
	F (−40 °C)	30	4.15 ± 0.03 ^e^*	10.26 ± 0.06 ^a^*	7.20 ± 0.02 ^d^*	7.12 ± 0.08 ^e^*
		90	4.09 ± 0.02 ^de^*	10.41 ± 0.07 ^a^*	7.09 ± 0.09 ^cd^*	7.00 ± 0.09 ^de^*
		180	3.82 ± 0.02 ^c^*	11.48 ± 0.05 ^b^*	6.36 ± 0.01 ^b^*	6.21 ± 0.02 ^c^*

Abbreviations are: FM, fresh milk; F, frozen milk; LAB, lactic acid bacteria; St, *Streptococcus thermophilus*; Lb, *Lactobacillus bulgaricus*. ^a–e^ Lowercase superscripts indicate significant differences (*p* < 0.05) between the mean values of various indicators of raw milk in fresh and different frozen storage groups as assessed by Duncan’s multiple comparison method. * Superscripts indicate significant differences (*p* < 0.05) among the same yogurt samples stored for 1 and 21 days as assessed by Duncan’s multiple comparison method.

**Table 4 foods-12-03223-t004:** Effects of frozen storage of yak raw milk on texture of yogurt.

Storage Time (Days)	Frozen Temperature	Frozen Time (Days)	Hardness (g)	Adhesiveness (g*s)	WHC (%)
1	FM	-	502.97 ± 15.70 ^f^	−755.85 ± 9.68 ^a^	46.00 ± 0.82 ^e^
	F (−20 °C)	30	476.26 ± 23.99 ^e^	−737.67 ± 11.00 ^a^	45.53 ± 0.73 ^e^
		90	369.87 ± 13.60 ^c^	−441.61 ± 13.31 ^c^	39.56 ± 0.73 ^b^
		180	261.18 ± 12.94 ^a^	−369.31 ± 15.88 ^e^	34.95 ± 0.69 ^a^
	F (−40 °C)	30	495.13 ± 11.25 ^f^	−740.05 ± 13.78 ^a^	46.53 ± 0.55 ^e^
		90	387.63 ± 9.99 ^d^	−480.56 ± 17.43 ^b^	43.00 ± 0.35 ^c^
		180	302.17 ± 11.07 ^b^	−395.46 ± 27.92 ^d^	40.77 ± 0.82 ^b^
21	FM	-	402.49 ± 4.87 ^e^*	−502.01 ± 20.80 ^a^*	41.23 ± 0.85 ^e^*
	F (−20 °C)	30	374.34 ± 14.99 ^d^*	−390.17 ± 15.81 ^c^*	37.43 ± 0.35 ^d^*
		90	279.82 ± 4.51 ^c^*	−260.74 ± 14.22 ^e^*	34.53 ± 0.15 ^b^*
		180	162.82 ± 6.39 ^a^*	−174.95 ± 14.04 ^g^*	32.07 ± 0.25 ^a^*
	F (−40 °C)	30	390.59 ± 8.32 ^e^*	−414.10 ± 20.85 ^b^*	39.27 ± 0.82 ^e^*
		90	289.37 ± 10.75 ^c^*	−330.15 ± 20.43 ^d^*	37.36 ± 0.32 ^d^*
		180	206.69 ± 14.88 ^b^*	−227.34 ± 23.77 ^f^*	35.43 ± 0.36 ^c^*

Abbreviations are: FM, fresh milk; F, frozen milk; WHC, water holding capacity. ^a–g^ Lowercase superscripts indicate significant differences (*p* < 0.05) between the mean values of various indicators of raw milk in fresh and different frozen storage groups as assessed by Duncan’s multiple comparison method. * Superscripts indicate significant differences (*p* < 0.05) among the same yogurt samples stored for 1 and 21 days as assessed by Duncan’s multiple comparison method.

## Data Availability

All data generated or analyzed during this study are included in this published article.

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
