# Peer review of "Effects of Freezing Raw Yak Milk on the Fermentation Performance and Storage Quality of Yogurt"

_foods, 2023, doi:10.3390/foods12173223_

Round 1

Reviewer 1 Report

Comments and Suggestions for Authors

The study of Li and colleagues touches an interesting topic in the dairy field. The manuscript is generally well written and the data clearly presented. However, some minor points needs to be addressed.

-line 12 - Here "fresh yogurt" should be substituted with "yogurt obtained using fresh milk";

-in all materials and methods the authors should specify the number of replicates they did for each quantification;

-Tabe 1- the authors should argument why the protein and the fat contents are decreasing during the storage at -20 and -40 °C. Is the freezing compatible with protein and lipid  hydrolysis? If this is the case, the authors should have measured an increase of peptides fatty acids and amino acids. In addition appropriate references should be cited.

- Figure 1 - Here, the acidification curves together with the yogurt cultures growth would have been much more informative. Could acidification data be added?

-Table 2 - The authors should specify how they have calculated the growth kinetic parameters (in which time intervals did the authors calculate the growth rate and V max?);

-Table 3 and the related text - Why the yogurt pH decreased during storage when yogurt was obtained using milk frozen at -20 °C and -40 °C? The lower pH measured is in contrast with the lower yogurt cultures cell count. The authors should comment on that.

-Figure 3. The authors should use the same scale of the Y axis in the two plots.

-line 390-391 - The authors wrote "the yogurt in the - 20 °C group had lower bacteria count than the industry standard". But the industry standard was ≥ 6 log CFU/mL. All the samples reach that value.

Author Response

See annex for details

Reviewer 2 Report

Comments and Suggestions for Authors

P2L79: fresh raw yak milk from the same was used as a control group for yogurt production: you mentioned that (Samples were collected in the farm and frozen at -80°C, flown to the laboratory) after thawing (and frozen in a refrigerator at -20°C and -40°C for 30, 90 and 180 days, respectively). More explanation is required regarding fresh raw yak milk from the same batch? How it is fresh and from the same batch, while you produced the yogurt for the other treatments after 30, 90 and 180 days.

P2L80: Samples from each group were homogenized (15 MPa) and pasteurized (62 °C, 30 min): In most cases, milk is first pasteurized and then homogenized to mix and disperse the milkfat throughout the milk to create a uniform mixture.

P2L81: 106 CFU/mL of yogurt starter MY105 (containing Lactobacillus bulgaricus and Streptococcus thermophilus in a 1:1 ratio, Danisco (China) Ltd) was added to the yak milk: What was the percentage of starter addition? As example 1%, 2% or 5%.

P2L87: (2.3.1 Milk composition, pH and acidity): should be before (2.2 Preparation of yak yogurt)

P2L92-98: During fermentation, the pH of the yogurt was measured every 30 minutes until pH reached 4.5. The fermented yogurt was refrigerated at 4°C and evaluated at 1 and 21 days Foods 2023, 12, x FOR PEER REVIEW 3 of 15 of storage. The results obtained were simulated using a modified Gompertz equation [19] of bacterial growth applicable to the pH drop during fermentation (Equation 1): I think this paragraph doesn’t match with the title (2.3.1 Milk composition, pH and acidity) and I suggest to put it under the previous title (2.2 Preparation of yak yogurt)

P3L99: (2.3.2 Protein hydrolysis degree of raw milk): should be before (2.2 Preparation of yak yogurt)

P3L107: (2.3.3 Thermal stability of raw milk): should be before (2.2 Preparation of yak yogurt)

P3L114: total psychrophile count (TPC): (TPC) stands for total plate count, while (TPsC) stands for total psychrophilic counts, Not (psychrophile). 

Comments on the Quality of English Language

Minor editing of English language required

Author Response

See annex for details

Round 2

Reviewer 1 Report

Comments and Suggestions for Authors

The revised manuscript was ameliorated compared to the original submission. However, some critical issued have not been addressed by the authors. Specifically the decrease in protein and fat content in milk stored at -20 and -40 °C must be explained. The hypothesis that this "phenomenon was caused by the presence of some active cryophilic bacteria during the freezing period " is not credible. I therefore suggest to delete these data and all the related comments.

The acidification curves should be all in the same plot to better understand similarities and differences between samples.

Concerning the data in Table 3. The justification written by the authors is not convincing. The lower pH measured some sample is in contrast with the lower yogurt cultures cell count. Are the acidification curves in agreement with the pH values measured?

Author Response

see annex
